# A Machine Learning Approach for Predicting Non-Suicidal Self-Injury in Young Adults

**DOI:** 10.3390/s22134790

**Published:** 2022-06-24

**Authors:** Pere Marti-Puig, Chiara Capra, Daniel Vega, Laia Llunas, Jordi Solé-Casals

**Affiliations:** 1Data and Signal Processing Group, University of Vic—Central University of Catalonia, 08500 Vic, Catalonia, Spain; pere.marti@uvic.cat (P.M.-P.); chiara.capra@uvic.cat (C.C.); 2beHIT, Carrer de Mata 1, 08004 Barcelona, Spain; laia.llunas@behit.cat; 3Psychiatry and Mental Health Department, Hospital Universitari d’Igualada, Consorci Sanitari de l’Anoia & Fundació Sanitària d’Igualada, 08700 Igualada, Barcelona, Spain; daniel.vega@uab.cat; 4Department of Psychiatry and Forensic Medicine, Institute of Neurosciences, Universitat Autònoma de Barcelona (UAB), 08193 Cerdanyola del Vallés, Barcelona, Spain

**Keywords:** NSSI, EMA, app, machine learning

## Abstract

Artificial intelligence techniques were explored to assess the ability to anticipate self-harming behaviour in the mental health context using a database collected by an app previously designed to record the emotional states and activities of a group of subjects exhibiting self-harm. Specifically, the Leave-One-Subject-Out technique was used to train classification trees with a maximum of five splits. The results show an accuracy of 84.78%, a sensitivity of 64.64% and a specificity of 85.53%. In addition, positive and negative predictive values were also obtained, with results of 14.48% and 98.47%, respectively. These results are in line with those reported in previous work using a multilevel mixed-effect regression analysis. The combination of apps and AI techniques is a powerful way to improve the tools to accompany and support the care and treatment of patients with this type of behaviour. These studies also guide the improvement of apps on the user side, simplifying and collecting more meaningful data, and on the therapist side, progressing in pathology treatments. Traditional therapy involves observing and reconstructing what had happened before episodes once they have occurred. This new generation of tools will make it possible to monitor the pathology more closely and to act preventively.

## 1. Introduction

Non-suicidal self-injury (NSSI) can be defined as the deliberate self-inflicted destruction of body tissue without direct and conscious suicidal intent [1,2]. It has been included as an independent disorder in the Diagnostic and Statistical Manual of Mental Disorders, Fifth Edition (DSM5), as a ‘condition requiring further study’ [3]. The incidence of NSSIs among adolescents and young people has increased significantly during recent years and has become an alarming problem [4,5]. Moreover, NSSIs have an impact on sufferers by compromising their academic performance [6] and generally increasing interpersonal difficulties, generating psychological distress and ultimately triggering actual suicide attempts [7]. According to the existing theories, NSSIs provide a means of regulating unpleasant emotions with the aim of regaining an equilibrium and returning to a balanced emotional state [8,9]. Consequently, individuals who practice NSSI tend to experience more unpleasant emotions, as found in self-report studies [10]. Self-report studies, the most common technique for studying motivations for NSSIs, suffer from the limitation of requiring subjects to remember reasons and motives, and are inherently biased due to their retrospective nature [11]. Although a significant proportion, of almost 30% of subjects, are unable to identify the reason for their NSSI, the most common response of subjects when asked about the reasons they engage in NSSI refer to a relief from unpleasant emotions and/or thoughts [12].

A technique that can uncover the motives for NSSIs is called Ecological Momentary Assessment (EMA). This technique makes it possible to explore the antecedents and consequences surrounding NSSI acts. Moreover, the data are collected in real time and in the context in which they occur [13]. This, in turn, avoids the bias introduced by trying to recall motives and mood retrospectively, and ecological validity is maintained [11]. However, there are few published studies using this technique to determine the motives that triggered an episode of NSSI [14,15,16]. Previous published work indicates that an increase in negative emotions is one of the common events prior to an episode of NSSI [17,18]. Therefore, it appears that negative affect is a strong predictor of NSSI.

In this work, we aim to introduce machine learning techniques on EMA data to investigate which emotions allow the system to detect new data as belonging to an NSSI or non-NSSI class. By developing such a system, the output of the model can be used to interact with the subject, personalising the messages that an app displays and with the aim of avoiding an NSSI episode. In addition, the model can also trigger a warning or an alarm to the psychologist or medical team supervising the subject, in real time, and deploy more active team actions in critical cases.

Machine learning (ML) methodologies, due to their significant advantages in automation, training, self-improvement, capacity to handle multi-variety data, and high prediction of model accuracy, have been penetrating the mental health field [19]. In addition, ML allows for quantifying with precision key features of the therapeutic process, which can be valuable for informing the therapy, thus allowing for the automatic assessment of treatment quality and outcome to predict performance at the individual patient level [20]. Nonetheless, research which focuses on developing algorithms to predict treatment response (TR) is very limited and uses a great variety of data, as well as ML methodologies which differ in nature. In our research, we have grouped previous works into three different groups, separating them basically by the type of data they use. Accordingly, we organised the related literature according to whether the data used are magnetic resonance imaging (MRI/fMRI) data, baseline patient data or conversation data.

Most mental health studies that apply ML techniques use MRI/fMRI data. Thus, by using MRI/fMRI data, the study by [21] predicted for the first time the individual outcomes of cognitive behavioural therapy (CBT) with a random forest, achieving 79% accuracy in predicting treatment response (TR). The same ML methodology was used in [22] to predict individualised treatment in borderline personality disorder (BPD) and reached 70.25% of accuracy. In [23], a support vector machine (SVM) was used to predict CBT outcomes in patients diagnosed with a social anxiety disorder (SAD) after one year with 92% accuracy, and [24], also with a SVM, predicted TR for patients with major depression disorder (MDD) by employing the response to faces showing different emotions. In [25], evidence was provided that symptoms of affective disorders were improved by predictions of enhanced neural activity when exposed to cognitive paradigms of psychosis by using a multivariate pattern analysis. The work in [26] found that a cross-validation model of functional connectivity (FC) patterns predicted single-subject post-TR better than clinical predictors alone. Among all the neuroimaging studies in the mental health field, the SVM methodology has been recognised as one of the most effective methods for making accurate single-patient predictions both in the short and long term.

On the other hand, baseline patient data are also valuable for personalising pre- and during-treatment recommendations to improve treatment and cost-effectiveness. For example, in [27], baseline features consisting of demographics, treatment type, neuropsychiatric questionnaires and clinical data were used to predict a single subject’s treatment outcome and patient costs before an online therapy for depression, compared to traditional therapy. The work in [28] provided evidence that trained ML models based on mere demographic and psychometric data were more effective (74%) in predicting long-term therapy outcomes for a patient undergoing CBT affected by alcohol dependence compared to clinical staff’s predictions (40–50%). A similar study in [29] used prognostic data of a sample of patients diagnosed with schizophrenia, successfully predicting symptom relapse through a generalised estimating equation (GEE) model and artificial neural networks (ANN).

Finally, we compiled a group of works that use conversation data. Conversational analytics within psychotherapy have also been recognised as a powerful tool for extracting specific data features to develop ML models for TR. For instance, Hoogeendorn extracted features from 69 individual text therapy sessions for social anxiety disorder (SAD) and applied sentiment analysis encompassing word frequency, writing style and conversations to predict TR [30]. The overall model predicted TR with 78% of specificity. A similar work applied NLP to a clinical dataset of 80,885 counselling conversations to predict ultimate therapeutic outcomes [31]. Likewise, in [32] about 14,899 CBT data points were analysed with deep learning to successfully predict TR measured as a decrease in significant clinical symptoms and therapeutic engagement. Finally, [33] is the only study where acoustic speech signal features, such as intensity and intonation, were recorded and analysed through NLP to predict the couple therapy outcome.

Although the literature in this field appears to be limited and heterogeneous in nature, the findings shed a light on the great advantage of ML applications in predicting TR and therefore predicting performance at the individual patient level. However, a series of major considerations need to be taken into account when applying ML to psychotherapy research. For instance, sample size in ML needs to be bigger than for traditional statistical methods, due to the fact that large datasets are needed to train and therefore fit the computational models. Furthermore, most of the studies found in the literature have used only computational approaches, while very few have compared ML methods with traditional statistical methods. Lastly, the willingness of subjects to fill in questionnaires or app fields, to allow access to data and metadata, together with an App’s ability to infer with the subject’s tasks is very relevant and needs to be taken into account [34]. Moreover, very few studies apply cross-validation methods, especially cross-culturally. Because of the difficulties in generalising results, the commercialisation of these clinical tools may be imperilled.

In our case, we analysed a database created through a mobile application. This contains data from both patients and controls. In the analysis, in addition to dealing with data whose classes are very unbalanced, we faced typical problems that are found in reality, such as dealing with incomplete data or having to deal with subjective assessments of each individual. In addition, found users who use the App a lot, generating many inputs, and others who use it very little, reporting much less information. For this reason, we selected extremely simple but easily interpretable classification systems, such as trees with only five splits. Thus, prior to the generation of functional apps, we must establish the ability of ML techniques to detect the positives and false positives obtained, and the study must be conducted at the individual level.

The rest of the paper is organised as follows: Section 2 contains the Materials and Methods, with a description of the participants, the app and EMA data, the preliminary work, and the full description of the machine learning methods used. Results and Discussion are in Section 3, detailing the new data visualisation introduced in our work, along with a discussion of the results obtained with the classification tree. Finally, Section 4 contains the Conclusions and points out some lines of future work to be developed.

## 2. Materials and Methods

### 2.1. Participants

The data used in our work are the same used in [16,35] and consist of 64 young adults (ages between 18 and 33 years), divided into three groups. The first two groups consist of participants with NSSI (≥5 NSSI events in the previous 12 months): (i) a subclinical group of university students recruited from the city of Igualada in Barcelona, Spain (STD group; N = 19), and (ii) a clinical group of BPD patients (BPD group; N = 22). The healthy control group consisted of 23 healthy participants (HC group), who were recruited via local advertisement and did not have any past or current mental disorders. All the participants completed self-report measures of borderline pathology, emotion dysregulation, de-centredness and negative emotional symptoms, and used the Sinjur application (EMA instrument) at least three times a day for 15 days to capture negative affect and NSSI in daily life. We refer the reader to [16] for further details on this matter.

### 2.2. Sinjur App

The Sinjur app was developed for the purpose of this kind of study, and was used to collect the data first analysed in [16]. We were not involved in its design, nor in the data collection step. The Sinjur app aims to help patients suffering from NSSI through a system based on cognitive behavioural therapy. The app collects EMA data based on the Experience Sampling Method (ESM), through three main sections, including emotions, daily activity and self-injuries. The app was designed to assess a variety of emotional states and dysfunctional behaviours. On each measurement occasion, participants were asked to indicate their momentary negative affect by (a) choosing from a list of emotions such as anger, guilt, frustration and sadness, and (b) rating the intensity of these emotions from 0 (lowest) to 100 (highest). They were also asked to indicate: (c) the number of times they were involved in arguments or fights with others, and (d) whether they had engaged in NSSI (yes/no) since the last measurement occasion. The app was configured to send participants reminder notifications 3 times a day to engage in the app and register data. A screenshot of the app can be seen in Figure 1. The Sinjur app is available for iPhone (https://apps.apple.com/es/app/sinjur/id1233186679 (accessed on 15 June 2022)) and Android (https://play.google.com/store/apps/details?id=com.bcn.tca.selfinjury (accessed on 15 June 2022)).

### 2.3. EMA Data

For this study, we focused on emotions only. More specifically, negative and positive affect were considered. As for the characteristics of the data, the collected data include a variety of demographic information (code id, sex, age, day, and the time at which the data are collected), together with the current emotions manifested by the subject through the app in the form of a percentage according to the level of emotion felt (0–100%). Many of these emotions are at 0% as usually the subject only adjust values for a few of them. Therefore, the data are sparse. The emotions to be reported by the users are listed in Table 1.

The next data block refers to the activities during the day. The activities available to report trough the app menus are: hanging out with friends, playing sports, working, listening to music, staying at home, studying, watching TV, surfing the Internet, other, being with family, eating, number of food binges, number of thoughts of self-harm, number of times taking drugs, number of times having sex and number or discussions with others.

Finally, there is a set of fields that the subject fills only in case of self-harm. Here the type of self-injury, the motive, and the feelings the subject had before and after the self-injury are collected. Again, the data for these last blocks are highly sparse because subjects were involved in one single activity and if they committed a self-injury, this is of one type only per case.

### 2.4. Model and Training

The model developed is conditioned by the typology of the data in different aspects. We took advantage of the self-injury records that capture the subject’s emotions before and after the self-injury to label the data as positive (1) if the sample reports emotions before and after the NSSI episode, or negative (0) if nothing is reported in this block. Having this information allowed us to develop a supervised model. The aim was to anticipate a self-injury act before it occurs, using the data collected by the application. Therefore, we used current emotions and activity during the day, prior to self-harm, which are available for all samples, regardless of whether he or she self-harms. By proceeding in this way, we managed to label all the data. In fact, we found that the emotion data overlap between classes and the task of separating the classes was not trivial.

User interactions with the app were uneven. The number of interactions recorded with mobile devices varied considerably between users. This variability is represented graphically in Figure 2, where we can see, for example, that subject 19 interacted 142 times, but subject 27 only interacted once. It should be noted that, during the experiment, 24 users reported one or more self-injuries, which is slightly more than a third of the total. In the figure, the set of messages/interactions that the subject linked to a self-injury are shown in red, while the interactions that did not trigger self-injury (or were not reported) are shown in blue. A total of 2144 entries were recorded, 78 of which report self-injury while the remaining 2066 entries are not linked to self-injury. The subjects who reported self-injury can be identified in Figure 2 because in their corresponding bar, there is a portion of red colour. Note that subjects who self-injure usually do so more than once, so that almost all of them have more than one red entry.

### 2.5. Leave-One-Subject-Out Cross Validation

The analysis of the model was carried out using leave-one-subject-out (LOSO) cross-validation. This means that a single subject is iteratively left out of the training dataset and used exclusively to test the model [36]: all the samples of one subject are kept out to create the model with the data of the other subjects, and then the model is tested on all the samples of the subject that was not in the modelling step. With a total of 64 subjects in our data, we derived 64 different models, each one with a different number of samples due to the varying number of daily reports made by each user. Thus, after training the model with all subjects except subject *j*, we tested the model using subject *j*’s data. As each subject may or may not have self-injured, in some cases the test data only contained samples from one class, but as 63 of the 64 subjects were in the training set, we ensured that we have examples from both classes in the training. In addition, to balance the training phase, we randomly selected samples from class 0 (no self-injury) to fit the same number of samples from class 1 (self-injury), as this was the minority class. We then modelled the system and tested it. To avoid biases due to the samples used from the majority class, we repeated this process 100 times and finally extracted the mean value of the confusion matrix of subject *j*. To summarise all models, a final mean confusion matrix of the 64 models is presented. From this, we calculated useful parameters such as Accuracy (Acc), Sensitivity (Sen), Specificity (Spe), Positive Predictive Value (PPV), Negative Predictive Value (NPV) and F1 score.

We want to emphasise the need for using LOSO as the evaluation methodology. As reported in [37], *k*-fold evaluation results show that if data from all subjects are used for training, the model tends to perform well. In this case, the training and test data are not separated in terms of subjects, and therefore, both the training and testing sets contain data from each subject, and hence a high level of accuracy can be obtained. The standard *k*-fold machine learning evaluation methodology learns subject-specific features rather than disease-related features, and because in a more realistic scenario the goal is to classify a new unseen subject, we report subject-independent results using LOSO. Because some subjects were difficult to classify, the standard deviation in the LOSO results is high. This effect could be hidden if we used *k*-fold instead of LOSO because these difficult subjects would be distributed across test sets, giving a false sensation of very good overall results.

### 2.6. Preliminary Work

In [35], we explored this dataset for the first time. Two classification trees were used as models. For the first tree, a coarse one with a maximum number of 4 splits, the training results showed 69.7% accuracy, whereas test results showed 59.3% accuracy. For the second tree, a larger tree with a maximum number of 20 splits, the training results showed 67.9% accuracy, whereas test results showed 65.2% accuracy. From this first attempt, two main difficulties where identified: (i) the classes are not easily separable, with some of the features clearly overlapping between classes, and (ii) there is a large imbalance between classes, with very few positives. After several tests, we found that simple models performed as well as or better than complex ones. The use of complex models (ANN, SVM) allowed us to achieve higher accuracy but an over-fitting effect appeared, and therefore the models were not general. However, simple models (coarse trees) generalised better. The other important information we gathered was that with three input features, we achieved the same performance as using all 12. The most important feature for classification was the number of self-harm thoughts. The two additional ones varied depending on the tree, with a preponderance of $f_{2}$ and $f_{5}$ (see Table 1). From these two findings together, we saw that the coarse classification tree was a very interesting option, because in addition to obtaining high performance compared to the other techniques, it provided an interpretation of the trigger of the self-injury.

The main limitation of the previous study is that, in order to capture a considerable proportion of positives, we also collected a very high number of false positives. These false positives are spread across the frames of all users and false alarms are generated in individuals who would presumably be at low risk. Although a false positive may be used to issue a warning, the generation of a very high proportion of false positives detracts from the effectiveness of the application.

### 2.7. Machine Learning Model

Tree-structured classification techniques have been widely used in medical applications. The reason for this is the ease of interpretation and applicability provided by these models. Therefore, the classification model used in this work will be based on a Classification and Regression Tree (CART) [38,39]. Following a similar strategy as in [35], we will use trees to derive our classification models.

A classification tree is a way of representing the knowledge obtained in an inductive learning process. It is a supervised classification method, which means that it uses already labelled data from which knowledge will be extracted. The feature space is subdivided by using a set of conditions, and the resulting structure is the tree. A tree consists of two different types of nodes, internal nodes and end nodes (also known as leaves). Each internal node solves a question about a particular feature *f* of the type “Is *f* greater than or equal to a threshold or not?”, and provides two children (subdivisions), one for each possible answer, depending on whether f≥ threshold or f< threshold. On the other hand, end nodes are those that are assigned to a single class at the bottom of the tree, so there are no further subdivisions from them. The construction process is recursive and starts by considering all possible partitions and choosing the one with the best separation. Then the optimal partitioning is applied, and the previous step is repeated for all the internal nodes [38,39]. A key point in this process is how the best separation is defined. In a general way, the best separation is the one that divides the data into groups such that there is a dominant class. To measure that, the algorithm in our experiments uses the Gini diversity index, which is one of the possible impurity measures [40]. The Gini diversity index is a measure of how often a randomly chosen item from a set would be incorrectly labelled if it were randomly labelled according to the distribution of labels in the subset. Gini impurity can be calculated by summing the probability of each item being chosen multiplied by the probability of an error in the categorization of that item. It reaches its minimum (zero) when all cases in the node correspond to a single target category.

Thus, the importance of the characteristics is established. The first-level characteristics are the most important. Similarly, the lower-level features are the less important ones. If the algorithm keeps some of the available features out of the tree definition, it means that these features are irrelevant to the classification model. This is one of the most interesting capabilities of trees, because it means that the model can be interpreted in terms of the features used in it and the features discarded by it. Hence, by analysing the structure of the tree, we can infer the interest of each of the chosen explanatory variables.

All the experiments in this work were carried out using Matlab and its machine learning toolbox. The classification tree was implemented using the fitctree function and the aforementioned Gini index, with a maximum of 5 splits allowed. To train the classification tree, the entire set of emotions and daily activities listed in Table 1 was used as the input. For each sample, the class label was obtained based on the presence or absence of self-harm in that sample. Class 1 corresponds to positive NSSI (self-harm reported in that sample), while class 0 corresponds to negative NSSI (no self-harm reported in that sample).

## 3. Results and Discussion

### 3.1. Visualization

The results and their interpretation will be presented through the confusion matrix obtained for each of the 64 users using the LOSO methodology. To visualise the results for all 64 users simultaneously, we adapt the typical confusion matrix representation as follows: (i) first, we calculate the percentage of true positives (TP), true negatives (TN), false positives (FP) and false negatives (FN), because for each subject there is a different number of cases; and (ii) we shift the confusion matrix to a bar representation in which the size of the partitions is proportional to the percentage of the matrix. In order to get an idea of the number of interactions of each user and the number of interactions that each percentage represents, we base the size of the bar on the size of interactions made by the subject. In addition, for ease of interpretation, a colour code is used. Figure 3 illustrates this way of representing the confusion matrices.

Thus, Figure 4 shows the results of the test. First of all, note that the profile of the bars is the same as the one depicted in Figure 2 describing the content of the labelled database, as expected. In this figure, note that most of the TNs, which form the most frequent case, are correctly identified, hence the predominance of the light green colour. Note also that the TPs appear in dark violet and correspond to most of the positives marked in red in the figure. This is therefore a good behaviour of the classifiers because it means that a very high proportion of the frames labelled as positive are being detected as such. However, some of the records labelled as positive are not detected correctly, i.e., they are FNs, which correspond to the dark green section. In this sense, the most critical cases would be those of subjects 5, 7, 15, 40, 49 and 58, in which self-harm is not detected despite having been present. What these six subjects have in common is that they have made low use of the app. They are all in the low range in terms of the number of reported entries. Finally, false positives (FP) are the orange areas in Figure 4. In some cases, these entries warn of suspected self-harm from frames/entries that were labelled as such, but in subjects who end up presenting self-harm, as in the case of subject 18, for example. In general, it is of interest that the rate of FPs is as low as possible, so that the app can provide positive feedback to the user without being too intrusive and only when it is really useful, not indiscriminately.

Given that the classes in the database are highly unbalanced, the system could maximise the accuracy by assigning all the cases to the majority class, but of course such a system would be useless. What we are looking for is a system able to detect the positives cases efficiently, maximising the probability of detecting them. Figure 5 represents the information of the database as it was labelled and keeps the colour code of Figure 2 together with the positive outputs (class 1) given by the system. Thus, on the negative part of the ordinate axis, for each user, the set of TPs and FPs provided by the system are depicted, following the colour convention established in Figure 4.

### 3.2. Performance

In order to provide an overall measure of the system, we averaged the confusion matrices obtained for each subject expressed as a percentage, and therefore independently of the number of interventions made in the application. We call this the average confusion matrix $\bar{CM}$ and it calculated as:(1)$$\bar{CM}=\frac{1}{N_{u}}\sum_{i=1}^{N_{u}}{CM}_{i}=\left[\begin{array}{cc} \bar{TN} & \bar{FP}\\ \bar{FN} & \bar{TP} \end{array}\right]=\left[\begin{array}{cc} 82.41\% & 13.94\%\\ 1.28\% & 2.36\% \end{array}\right],$$
where ${CM}_{i}$ is the confusion matrix of subject *i* expressed as percentages and $N_{u}$ the number of users. $\bar{TN}$, $\bar{FP}$, $\bar{FN}$ and $\bar{TP}$ are the mean values of TN, FP, FN and TP, respectively, also expressed as percentages. This aggregation allows us to define quality measures for the classifier. One of the important measures will be the precision or positive predictive value (PPV) and the negative predictive value (NPV) that give us an idea of the probability that a positive provided by the classifier is really a true positive and a negative is a true negative, respectively. We obtain them from the following expressions: (2)$$PPV=\frac{\bar{TP}}{\bar{TP}+\bar{FP}}=0.1448$$
(3)$$NPV=\frac{\bar{TN}}{\bar{TN}+\bar{FN}}=0.9847$$

In the same way we calculate the Accuracy, Sensitivity and Specificity: (4)$$Accuracy=\frac{\bar{TN}+\bar{TP}}{\bar{TN}+\bar{TP}+\bar{FN}+\bar{FP}}=0.8478$$
(5)$$Sensitivity=\frac{\bar{TP}}{\bar{TP}+\bar{FN}}=0.6484$$
(6)$$Specificity=\frac{\bar{TN}}{\bar{TN}+\bar{FP}}=0.8553$$

We can see that the probability of correctly predicting a negative is very high, 98.47%, due to its predominance in the whole database, but that the probability of correctly predicting a positive is considerably lower, 14.48%. Since the Accuracy indicates the proportion of correct classifications, when numbers of observations in different classes vary greatly, as in this case, Accuracy alone could yield misleading results, so this parameter is usually accompanied by Sensitivity and Specificity, and especially by PPV. The obtained value of 14.48% for the PPV is a good result, especially if we compare it with the value of 4.58% obtained with the best classifier used in [35] working with this same database.

### 3.3. Discussion

The database used in this work had been used before in [16], in which a multilevel mixed-effect regression analysis was performed. This analysis showed that momentary frustration alone directly predicted NSSI. In our work, we followed a different approach based on classification trees. The feature that provided the most information for the classification of the instances were $f_{9}$ (self-harm thoughts), $f_{2}$ (sadness level), and $f_{7}$ (frustration level) (see Table 1), which appear in the vast majority of the generated models, and which is close to the results described in [16]. It is interesting to note that the more times the subject reports having thought about self-harm, the greater the likelihood of self-harm was. This confirmation may give hints on how to improve the application in the future by thinking about and implementing direct or indirect ways of collecting this parameter.

The proposed classifier is based on extremely simple trees, limited to five branches. In the experiments conducted, we found that with the collected database we achieved the most robust results using classifiers with only three input features. It is important to note that the database used for this study had many empty fields, with many empty features, as the subjects did not consider filling them in. In fact, subjects rarely filled in all fields. Given this observation, a simplification of the mobile application for data collection should be considered. For example, nine emotions are currently collected (Figure 1, left) and nine buttons (potentiometers) are used for this purpose (Figure 1, right), one for each emotion. We propose to reduce them, for example to three, where each button (slider bar or potentiometer) would collect one emotion and its opposite (e.g. happy–unhappy), so that the subject could fill them all in very quickly on the same screen. This improvement could lead to more complete and efficient data collection, and could avoid many empty fields, resulting in a less sparse database. With such a database, more refined models could improve the prediction results, being more beneficial for the user and the psychologists/medical team.

In a previous study [16], self-reports and momentary measures were combined. Of the momentary measures, the focus was put on momentary negative affect and on interpersonal conflicts, as triggers of NSSI episodes. However, thoughts of self-harm were not included. Expanding these previous findings, in the current work we included thoughts of self-harm in addition to other momentary measures. Interestingly, we found that thoughts of self-harm constitute a very relevant trigger of NSSI in the sample of study. This finding is partially consistent with the previous study, observing that a greater decentering ability predicted a reduced likelihood of NSSI engagement. This result indicates that the ability to observe one’s thoughts and emotions in a detached manner, as if they were transient events of the mind, have a protective role in NSSI. In contrast, higher rumination, the conceptual opposite of decentering [41], enhances or exacerbates the effect of negative affectivity on the likelihood of NSSI engagement [42]. In the current study, we observe that repetitive thinking about self-harm increases the likelihood of engaging in NSSI, which is consistent with the above. Our finding also complement previous findings with momentary measures, highlighting the role of NSSI thoughts in the engagement of NSSI. For instance, a recent study with adolescents and young adults showed that intense NSSI thoughts over prolonged time periods may deplete the self-regulatory resources required to terminate NSSI episodes once they have begun, leading to NSSI [43]. In this regard, future studies need to explore a possible interaction between self-reported decentering ability and momentary repetitive thinking about self-harm in young adults who engage in NSSI. Given that thoughts about NSSI could trigger NSSI acts, psychological interventions that improve decentering capacity (e.g., mindfulness [44]) could be helpful in avoiding NSSI.

This study suffers from some limitations that should be addressed in future work. First, the size of the database is not large. More data should be collected to confirm and improve the actual results. In addition, gender is not considered in this study, although it could be a variable influencing the results. The lack of this information does not allow this possible bias to be checked. As for the classification models, here we have only considered simple classification trees to facilitate the interpretation of the results. Other possible models could be explored, such as ensembles of decision trees, neural networks, etc. Furthermore, to improve the results, feature selection techniques could be investigated in the future. Finally, the fact that some fields are not filled in and are therefore considered as 0 in the model may also introduce a bias in the results. The use of the proposed new slider bars or potentiometers that capture an emotion and its opposite could be a solution for avoiding empty fields.

## 4. Conclusions

Our findings point out the relevance of thoughts as a proximal (short-term) risk factor for NSSI engagement. This result complements previous findings indicating that a greater capacity for de-centredness (i.e., the ability to observe one’s thoughts and emotions in an independent manner) predicted a lower likelihood of NSSI engagement [45]. It seems clear that the use of mobile applications can be of great help in the prediction of self-harm, as there are indications that data collected by an application running on a mobile device, properly processed, is able to anticipate self-harm. This anticipation can trigger some kind of action that can be delivered to the subject via the mobile device itself, as well as capturing valuable clinical information. Conducting studies such as this allows us to obtain objective metrics and, in collaboration with specialists in the field, to understand the triggering mechanisms of self-harm at an early stage. Identifying the moment that anticipates a self-injury would make it possible to initiate some kind of action to help the subject to inhibit the risk behaviour. In addition, this study allowed us to detect aspects that could be improved in the app’s data collection process, providing support tools in a health field with a high social impact. It is important to note that NSSI is a strong predictor of suicide, which is a public health priority. In this sense, the present results may have implications for suicide prevention in young adults. Our findings open the door to an Ecological Momentary Intervention (EMI) aimed at treating people who engage in NSSI during their daily lives and in natural settings, probably optimising the current psychological treatments available for NSSI. Importantly, our findings are independent of BPD, suggesting that assessment and subsequent intervention regarding NSSI thoughts may be a relevant treatment goal in both clinical and subclinical settings.

Nonetheless, these promising data need to be built upon in future studies and need major translation into the clinical field to demonstrate its real-world efficacy and, later, to be translated into the world of enterprise.

## Figures and Tables

**Figure 1 sensors-22-04790-f001:**
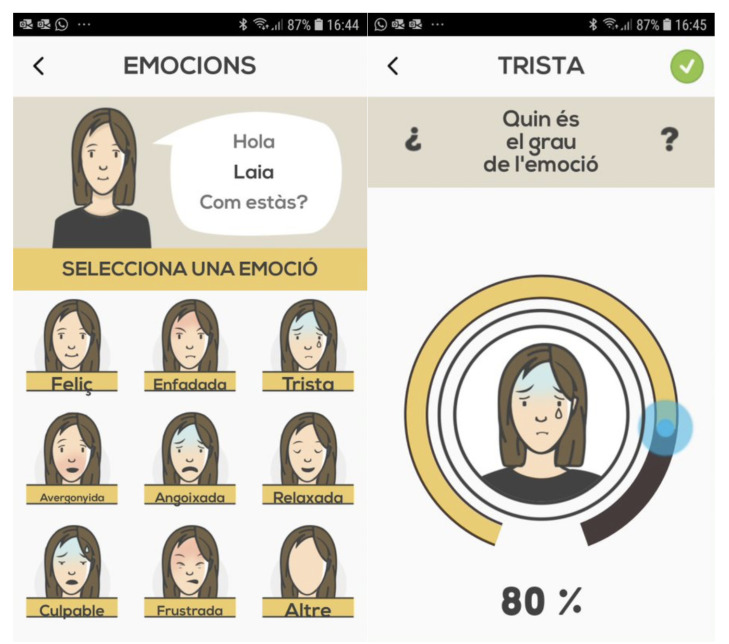
On the **left**, a screenshot with the list of emotions to report to the system. On the **right**, a detail on how to report the grade of the emotion by means of a sliding button (the text in the App is in Catalan. See the translated names of the emotions in Table 1).

**Figure 2 sensors-22-04790-f002:**
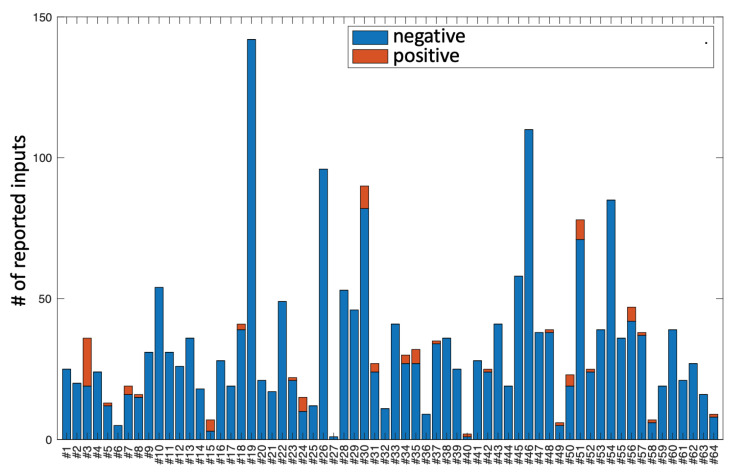
Representation of the number of interactions that each subject in the study had with his/her application. The proportion of these messages that were not directly linked to any self-harm is shown in blue, and that which was directly linked to self-harm in red. Note that of the 64 subjects present, 24 reported one or more self-injuries.

**Figure 3 sensors-22-04790-f003:**
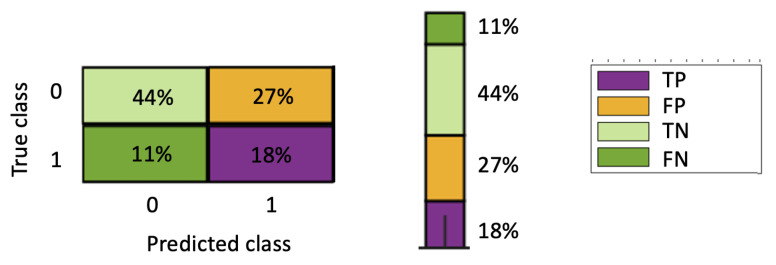
Representative example of the confusion matrix with bar shapes, used for the joint representation of all users’ data. The sections in the vector are proportional to the percentage of the cell corresponding to the confusion matrix. The height of the bar will be proportional to the number of interactions made by the subject. Note the colour coding to represent the TP (True Positives), the FP (False Positives), the TN (True Negatives) and the FN (False Negatives).

**Figure 4 sensors-22-04790-f004:**
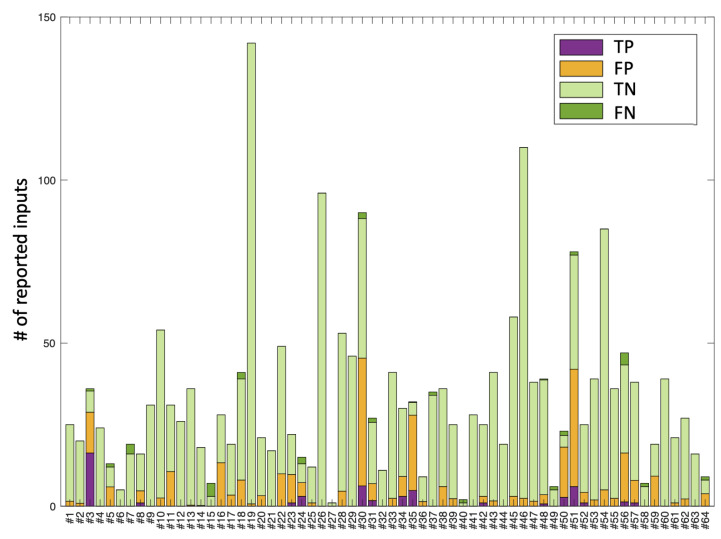
Representation of the results of the Leave-One-Subject-Out Cross-Validation test in terms of the vectorised confusion matrices scaled by the proportion of entries each subject made in the application. The vertical axis represents the number of entries for each user. We use the colour convention from Figure 3 to represent the TP (True Positives), the FP (False Positives), the TN (True Negatives) and the FN (False Negatives).

**Figure 5 sensors-22-04790-f005:**
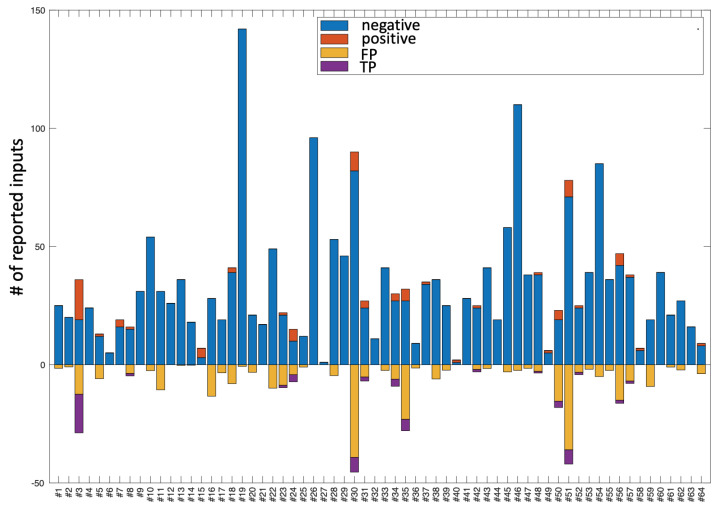
The positive part of the graph corresponds to the representation of the database in terms of the number of entries labelled as positive and negative for each user. The negative part of the graph corresponds to the TPs and TNs obtained in the Leave-One-Subject-Out Cross-Validation test.

**Table 1 sensors-22-04790-t001:** List of emotions and daily activities collected through the Sinjur app, together with the associated code.

Emotion (Original Name in Catalan)	Feature Code
Happy (*feliç*)	$f_{1}$
Sad (*trista*)	$f_{2}$
Embarrassed (*avergonyida*)	$f_{3}$
Distressed (*angoixada*)	$f_{4}$
Relaxed (*relaxada*)	$f_{5}$
Guilty (*culpable*)	$f_{6}$
Frustrated (*frustrada*)	$f_{7}$
# binge eating	$f_{8}$
# self-harm thoughts	$f_{9}$
# times taking drugs	$f_{10}$
# times having sex	$f_{11}$
# arguing with others	$f_{12}$

## Data Availability

Data sharing not applicable—no new data generated.

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
