# Peer review of "A Machine Learning Approach for Predicting Non-Suicidal Self-Injury in Young Adults"

_sensors, 2022, doi:10.3390/s22134790_

Round 1
Reviewer 1 Report
Summary:
In the paper titled "A machine learning approach to predict Non-suicidal Self-injury in young adults", the authors evaluated a so-called Ecological Momentary Assessment app that helps young adults who self-harm with non-suicidal intent. The evaluation is based on machine learning and aims to answer the question of whether it is possible to predict if a person will self-harm or not. The authors described their motivation and the medical problem. Then, the setting for the evaluation is described, consisting of the app, machine learning and the overall approach followed. Based on this, the results are discussed and limitations are described. The authors conclude the results of the paper with the conclusion that good indicators have already been found, as machine learning and EMA are helpful in this context, but that more data needs to be collected.
Points in favor of the paper:
- The paper is well written
- The paper has a clear contribution
- The paper shows experimental results
- The paper discusses related works
- The paper fits the scope of the journal
- The paper is factually sound
- The title of the paper fits the content
Points against the paper:
Although I generally like the paper, there are still some aspects to be addressed:
- The abstract needs to be rewritten, too few results are presented and also the content is not summarized well enough.
- Also the introduction needs to be improved, which machine learning was used, what should be predicted, where are the problems;
the reader learns many things too late
- Generally, in machine learning much more information has to be given, which methods exactly, not trees in general, what are
the backgrounds, which tooling was used, etc?
- the idea of EMA to use LOSO instead of k-fold is well founded, but it needs more info, is it trained on assessment level or
on user level. if the former is the case, then a user can appear again in the test and training set; but that depends on the
tooling as well
- it should be explained that there can also be a selection bias in the system, especially if there is so little data, and also if,
for example, there were only women.
- Furthermore there can be an information bias, line 91: "(0) if nothing is reported in this block" -> is this really a 0 or was
nothing filled in?
- There is very little said about the app, also there is no information bias in it
- in table 1 there is talk about features, where nothing was said about machine learning yet
- the achieved results should be connected to a baseline
- Commas should be checked in general
- Line 183, Page 6: "dily activities" -> "daily activities"
- not precise enough "In our experiments, the input features are the set of emotions and dily activities listed 183
in Table 1, and the two classes are the following ones: class 1 (self-harm, positive NSSI) and 184
class zero (no self-harm, negative NSSI)."
- Too few related works are discussed, as an example for the overall context:
https://link.springer.com/article/10.1007/s12652-019-01355-6
Author Response
We would like to thank all reviewers for their comments, which have provided the basis for improving the manuscript. We have thoroughly revised the manuscript according to the reviewers' comments and suggestions. We hope that the manuscript is now better and more interesting for the readers of Sensors.
In the attached PDF document, we provide point-by-point responses to each of the reviewers' comments and suggestions, with detailed descriptions of how the issues have been addressed. Please note that we have also reviewed grammatical errors and other issues not raised by the reviewers. We have improved the English language and style and corrected several typos and misused English terms.

Reviewer 2 Report
In my opinion the paper is not qualified for publication in Sensors Journal due to technical quality of the study.
The developed application App is not well described. For example, how are personalized the messages that the App displays.
The machine learning methods are not described in detail, and the overall study contribution is minimal.
The dataset is very unbalanced and could affect the results.
A discussion of the results obtained in the previous studies compared with those in this study is missing.
Author Response

(The authors gave the same response as above.)

Reviewer 3 Report
Thank you for the opportunity to review this interesting article. by Pere Marti-Puig and colleagues.
The paper aims to introduce machine learning techniques on EMA data and to develop a supervised model for anticipating a self-injury act before it occurs, based on the analysis of the collected data.
The topic is interesting and relevant and the article is well-written. The language used is concise and I like how the authors have shorten the text but they clearly explain every aspect of the research. The methodological approach and findings are clearly described, and the structure of the paper has a clear and logical flow. The findings and future work are also addressed.
I have some suggestions/concerns:
1. In my opinion, the novelty/originally of the paper is not clearly underlined. That is why, I would recommend to the authors to describe/specify if ML techniques are used, for similar approaches, in other research papers in field;
2. Comments concerning inaccuracies within the text:
a) Related to acronyms:
The DSM5 (Diagnostic and Statistical Manual of Mental Disorders, Fifth Edition) acronym is not defined (line 5);
The BDP (Borderline Personality Disorder) acronym is not defined (lines 59 and 286);
The CART (Classification and Regression Tree) acronym is not defined (line 161).
b) Ambiguity introduced into text: “would collect one emotion and its opposite (e.g. happy-happy)” (line 260).
I hope my feedback is useful to the authors in improving their paper and wish them all the best in pursuing this important area of research.
Author Response

(The authors gave the same response as above.)

Reviewer 4 Report
The article deals with an important topic, I consider it current and I recommend to the authors the expansion of the study and further research.
The methodology is well chosen and clearly explained. The results are clearly presented. However, the article has two major weaknesses that will need to be addressed:
· Literary research is weak, the authors have failed to cover the topic from a theoretical point of view and in terms of analysis of existing research. The authors need to strengthen the literary base with additional resources.
· The discussion of the results is related to this point. It will be necessary to include an adjustment of the literature search in the part of the discussion.
· I recommend the authors to explain how the sample is selected, given that the sample is not robust and the results only apply to a small group of people. This is a major limitation of the study.
Author Response

(The authors gave the same response as above.)

Round 2
Reviewer 1 Report
My concerns have been well addressed
Author Response
We are very grateful for your comments/suggestions which have helped us to improve the quality of the article.
Reviewer 2 Report
The authors carefully clarified the issues I raised in connection with their study, but the methodology and the weak results cannot be overlooked.​​​​​​​
To improve the results, a technique for feature selection and methods based on ensembles of decision trees should be considered in the future.
Author Response
We are very grateful for your comments/suggestions which have helped us to improve the quality of the article. The following paragraph has been added at the end of the discussion section to highlight the importance of your comment:
As for classification models, here we have only considered simple classification trees to facilitate the interpretation of the results. Other possible models could be explored, such as ensembles of decision trees, neural networks, etc. Furthermore, to improve the results, feature selection techniques could be investigated in the future.
Reviewer 3 Report
I consider that the authors have addressed my recommendations and the updated manuscript shows significant improvements and is more legible and interesting now.
I have only some minor request for the authors:
- As stated at https://www.mdpi.com/journal/sensors/instructions#preparation, “The abstract should be a total of about 200 words maximum”. The Content of the Abstract is appropriated, but its lengths is 287 words that exceeds the maximum admitted one. That is why, I recommend its reduction to 200 words.
- - In lines 120-121: the text “It contains both patients and controls” might be changed to “It contains both patients’ data and controls”.
Author Response
We are very grateful for your comments/suggestions which have helped us to improve the quality of the article.
We have adjusted the abstract to 198 words, to meet the journal's requirements. We have also corrected the typo in lines 120-121.
Reviewer 4 Report
The authors have improved the article. My requests took into account or explained the relevant reasons. I recommend publishing it.
Author Response

(The authors gave the same response as above.)
